# Ingestion of *Bifidobacterium longum* changes miRNA levels in the brains of mice

**Laura DeVries**[☯], **Cara Horstman**[¤a☯], **Marie Fossell**[¤b], **Clayton Carlson**[iD]*

Department of Biology, Trinity Christian College, Palos Heights, Illinois, United States of America

☯ These authors contributed equally to this work.
¤a Current address: Student Life Department, Crown College, St. Bonifacius, Minnesota, United States of America
¤b Current address: Department of Nursing, Marquette University, Milwaukee, Wisconsin, United States of America
* clayton.carlson@trnty.edu

**Data Availability Statement:** All relevant data are within the manuscript and its Supporting Information files.

**Funding:** The study was supported by the Trinity Christian College with Honors awarded to Laura DeVries and Cara Horstman. The funders had no

## Abstract

The purpose of this research is to investigate the relationship between the microbiota of the gastrointestinal (GI) system and relative gene expression of miRNAs and mRNAs in the brain. C57BL/6 mice and Balb/c mice are fed *Bifidobacterium longum*, a well-characterized probiotic bacterial species shown to change behavior and improve sociability of Balb/c mice. After feeding, RNA was extracted from whole brains and PCR arrays were utilized to determine changes in the gene expression of brain-specific miRNAs. The results of these PCR arrays reveal that the relative gene expression of mmu-mir-652-3p is sensitive to *B. longum* probiotic treatment in C57BL/6 mice. qPCR was performed to measure expression of Dab1, an mRNA target of this miRNA. Dab1 expression is also dependent on *B. longum*. The goal of this study is to further understand the relationship between the gut microbiota and its impacts on neurological gene expression and brain function.

## Introduction

The Microbiota-Gut-Brain axis is the bidirectional relationship between microbes living in the gut and the brain of their host [1]. Nearly 3,000 different prokaryotic species have been identified from human fecal material [2, 3]. Beyond their involvement in digestive health, these bacteria have been shown to be able to influence health, neurodevelopment, and behavior [4]. Experiments in rodents demonstrate that the gut microbiota can interact with the brain through numerous mechanisms including small molecules, the immune system, and the enteric nervous system.

Studies of blood metabolites in germ free mouse models have shown that the gut microbiota produces a number of important physiologically active metabolites [5–7]. Among these are short-chain fatty acids. Short-chain fatty acids produced by gut microbes include acetate, propionate, and butyrate. Changes to the fecal concentration of these molecules are associated with numerous neurological conditions including Parkinson's disease and Autism Spectrum Disorder (ASD) [8].

Additionally, the gut microbiota interacts constantly with the host immune system. The neurological consequences of this interaction is an active area of research, but it is thought that

role in study design, data collection and analysis, decision to publish, or preparation of the manuscript.

**Competing interests:** The authors have declared that no competing interests exist.

the gut microbiota may regulate neuroinflammation through disruption of monocyte recruitment [9]. Likewise, the adaptive immune system has key roles in both the brain and the gut microbiota and the makeup of the gut microbiota seems to impact the activity of the adaptive immune system in the brain [10, 11].

The gut microbiota also communicates with the nervous system directly. Metabolites made by the gut microbiota including tryptophan metabolites, serotonin, GABA, and catecholamines which are all able to activate the autonomic nervous system and influence brain activity through the vagus nerve [12]. These metabolites also stimulate and shape the enteric nervous system, the complex web of neurons that largely controls the gastrointestinal tract. Interestingly, many disorders associated with the brain also display changes to the enteric nervous system including Parkinson's disease, ASD, and Alzheimer's [13].

While it may not be surprising considering the number of connections in the microbiota-gut-brain axis, changes to the gut microbiota impact a wide range of behaviors including food intake, stress behavior, sociality and more, though much work still needs to be done [14]. For each of these behavioral changes, the complete mechanistic pathway from bacterial population to behavioral change has yet to be determined, but the study of so called psychobiotics continues to become clearer.

One example involves treating mice exposed to chronic stress with a combination of two psychobiotics, *Lactobacillus helveticus* and *Bifidobacterium longum* [15]. Mice treated with these bacteria showed changes in the expression of key transcription factors in the brain and increases in neurogenesis and brain plasticity as compared to mice that received a control bacteria species. These bacteria seem to help regulate stress response [15].

Balb/c mice are inherently anxious and display behavioral differences when compared to the C57BL/6 strain [16]. Changing the gut microbiota can impact behavior of Balb/c mice. For example, six week treatment with either of two species of *Bifidobacterium* reduced stress behaviors in Balb/c mice [17]. In addition to reducing overall stress behaviors, *B. longum* showed a positive role in the Balb/c mouse response to acute stress and depression.

These results indicate that psychobiotics like *Bifidobacterium longum* can improve behavior in Balb/c mice, but many steps in the mechanistic pathway between the microbe and the behavior remain a mystery. For example, while the gut bacteria is known to regulate the expression of miRNAs in the brains of germ free mouse models [18], the impact of *B. longum* on gene expression of miRNAs in the brain of Balb/c mice has not yet been explored.

In the current study, Balb/c and C57BL/6 mice were treated with *B. longum* and the expression of a set of brain miRNAs was measured. We have identified differences in the expression of a miRNA involved in neurological development and a corresponding deviation in the expression of an mRNA that shapes brain structure that is targeted by this miRNA.

## Methods

Our work involved sacrifice of mice by cervical dislocation. This project was prospectively reviewed, approved, and then observed by the Trinity Christian College Animal Research Advisory Group. We do not have a permit number.

## Bacterial growth

*Bifidobacterium longum* subsp. *longum* Reuter (ATCC 15707) was grown anaerobically in Sigma-Aldrich MRS broth with Tween 80 and L-cysteine and frozen in 100μL aliquots representing $10^9$ cells per tube. Tubes of MRS media with Tween 80 and L-cysteine were also frozen for controls.

## Mice

Twenty-one day old C57BL/6 and Balb/c mice strains were obtained from Charles River Laboratories. The mice were divided into the following four categories according to their strain and type of bacterial feeding: C57BL/6—Control, C57BL/6—*Bifidobacterium*, Balb/c—Control, and Balb/c—*Bifidobacterium*. There were four mice in each group for a total of 16 animals. The animals were kept in cages in a climate controlled environment with 12hrs of light and 12hrs of darkness. Mice in each experimental condition were kept in a cage together. Four cages with four mice each. This is the minimum number of animals we were comfortable using in this experiment. Once daily, the mice were fed either 100 μL of $10^9$ *B. longum* or 100μL of media. The order in which the mice were fed, and the placement of the cages was rotated daily. The presence of *B. longum* in the bowel was not measured for any mice with or without treatment. All researchers were aware of which mice were receiving bacteria and which were receiving media. After 22 days of bacterial feeding, the mice were euthanized by cervical dislocation and death was assessed before brains were harvested. All four mice, from all four conditions (sixteen in total) were euthanized. No mice were excluded from the collection and RNA extraction because no criteria were determined *a priori* for their exclusion.

## RNA preparation and collection

Individual brains were placed in Qiagen RNA*later* RNA Stabilization Reagent to stop gene expression. The RNA of each whole brain was harvested using a Qiagen RNeasy Mini Kit following the vendor's instructions.

## cDNA synthesis

cDNA was synthesized for each mouse brain using Qiagen miScript II RT Kit following the vendor's instructions for miRNA enrichment. No mice were excluded from the cDNA synthesis, so four mice from each condition, sixteen total, were synthesized.

## Gene expression–PCR arrays

MicroRNAs that are involved in neurological development and disease were measured for each mouse group using the miScript miRNA PCR Array Mouse Neurological Development and Disease (MIMM 107ZD-12). Using the miScript miRNA PCR Array, 12.5 μL of each cDNA sample within a mouse group was added to the same PCR Master Mix. cDNA from the four mice in the same group were pooled. The Master Mix was thoroughly mixed with a pipette and 25 μL was pipetted into each individual well of the miScript miRNA PCR Array. The plate was run according to the Qiagen's instructions. Briefly, the plate was run at 95˚C for 15 minutes then cycled through 40 rounds of 94˚C for 15 seconds, 55˚C for 30 seconds, and 70˚C for 30 seconds. Each treatment group (of four pooled samples) was assayed four times yielding four technical replicates of the pooled samples. Data analysis was completed using the Qiagen Data Analysis Centre with normalization by automatic selection from the full plate.

## Gene expression–miRNA qPCR and mRNA qPCR

Based on the changes of gene expression found in the Neurological qPCR arrays, a gene of interest, miR-652, was investigated based on its association with autism spectrum disorder. A more nuanced measurement of this gene's expression was performed using each biological replicate. Isolated RNA from each mouse replicate was reverse transcribed to form miRNA-enriched cDNA using Qiagen miScript II RT Kit and the manufacturer's instructions.

The expression of mmu-mir-652-3p was measured using Qiagen 10x miScript Mm_miR-652_3 (MS00012425) primer assay. qPCR was completed using QuantiTect SYBR Green PCR Master Mix according to the vendor's instructions. Samples were compared to a control primer assay (RNU6).

In order to measure expression of an mRNA target of mmu-mir-652-3p, isolated RNA from each mouse brain replicate was reverse transcribed to form mRNA-enriched cDNA using Qiagen miScript II RT Kit and the manufacturer's instructions. The mRNA-enriched cDNA was measured by qPCR using QuantiTect SYBR Green PCR Master Mix. No mice were excluded from these experiments. There were four biological replicates and four technical replicates for each reaction.

## Results, discussion, conclusions

### *B. longum* changes expression of miRNAs in the brain

To determine the effect of the gut microbiota on expression of miRNAs in the brain of mice, two different strains of mice were fed approximately one billion *B. longum* daily for three weeks. The expression of brain miRNAs for C57BL/6 and Balb/c mice was measured using Qiagen PCR arrays. These arrays test the expression of miRNAs involved in neurological development and health.

The two strains of mice show similar levels of expression over a set of 80 different important miRNAs (Fig 1), but, as expected, numerous miRNAs are expressed differently in the two

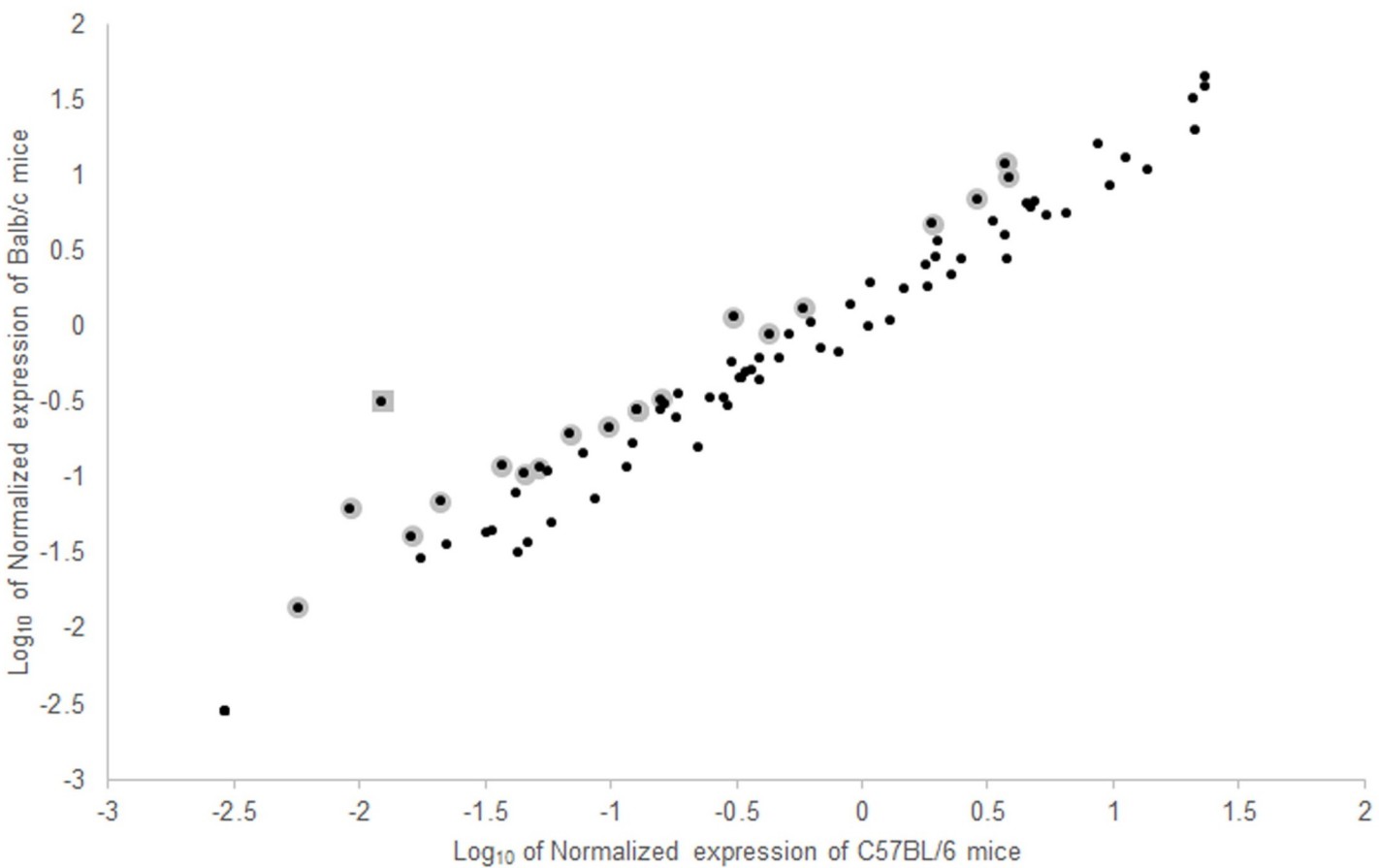

**Fig 1. Balb/c and C57BL/6 mice express some brain miRNAs differently.** When 84 miRNAs were tested using a PCR array, many miRNAs are expressed similarly in both strains of mice. However, some genes show differences expression in Balb/c as compared to C57BL/6 mice. Those with greater than 2-fold difference in expression are highlighted in grey. mmu-mir-652-3p is marked with a square (see below).

**Table 1. List of genes with more than 2-fold difference in expression levels in Balb/c and C57BL/6 mice.**

| Gene | Balb/c relative expression | C57BL/6 relative expression | Balb/c Fold expression over C57BL/6 |
|---|---|---|---|
| mmu-mmu-mir-652-3p | 0.317 | 0.012 | 25.72 |
| mmu-miR-33-5p | 1.152 | 0.310 | 3.72 |
| mmu-miR-328-3p | 0.258 | 0.076 | 3.4 |
| mmu-miR-431-5p | 0.119 | 0.037 | 3.22 |
| mmu-miR-26b-5p | 11.988 | 3.755 | 3.19 |
| mmu-miR-320-3p | 0.194 | 0.069 | 2.8 |
| mmu-miR-30e-5p | 9.670 | 3.834 | 2.52 |
| mmu-miR-338-3p | 4.769 | 1.904 | 2.51 |
| mmu-miR-138-5p | 6.885 | 2.865 | 2.4 |
| mmu-miR-485-5p | 0.105 | 0.045 | 2.32 |
| mmu-miR-181a-5p | 1.314 | 0.586 | 2.24 |
| mmu-miR-20b-5p | 0.115 | 0.052 | 2.21 |
| mmu-miR-146b-5p | 0.278 | 0.128 | 2.18 |
| mmu-miR-433-3p | 0.214 | 0.099 | 2.17 |
| mmu-miR-181c-5p | 0.885 | 0.426 | 2.08 |
| mmu-miR-107-3p | 0.328 | 0.159 | 2.06 |

strains of mice. miRNAs expressed more strongly in Balb/c mice than in C57BL/6 mice are shown in Table 1. These data suggest that there are differences in expression of regulatory RNAs in the brain in addition to the reported behavioral differences between these two strains of mice published previously [16].

In order to determine whether, and to what extent, short term probiotic usage could influence miRNA expression in the brain, Balb/c and C57BL/6 mice were fed approximately $10^9$ *B. longum* daily for three weeks. In both Balb/c and C57BL/6 mice, three weeks of daily feeding of the probiotic led to differences in expression of miRNAs that are important for neurological development and health (Fig 2). miRNAs that show at least 1.5-fold change in expression level in both strains of mice are reported in Table 2. This supports previous work that has shown that gene expression in the brain is influenced by the gut microbiota through the microbiota-gut-brain axis [18].

## *B. longum* reduces differences in miRNA levels in the brain of Balb/c and C57BL/6 mice

In order to determine whether the changes in miRNA expression associated with *B. longum* feeding can normalize expression between Balb/c and C57BL/6 mice, the level of miRNAs in mice of both strains after treatment with the probiotic were compared. After three weeks of treatment, Balb/c and C57BL/6 mice expressed brain miRNAs very similarly. Many miRNAs that are expressed differently when comparing the two strains of mice without probiotics show substantially more similar gene expression when treated with the probiotic (Fig 3). Most of the changes occur in the C57BL/6 mice. Of particular interest is the miRNA mmu-mir-652-3p, a regulatory RNA that has been implicated in human neurodevelopmental conditions like ASD [19]. According to the array data, Balb/c mice without *B. longum* treatment express mmu-mir-652-3p at much higher levels than C57BL/6 mice. However, with the probiotic, the levels of mmu-mir-652-3p is essentially indistinguishable between Balb/c and C57BL/6 mice. Both mice seem to display the higher level of gene expression for mmu-mir-652-3p.

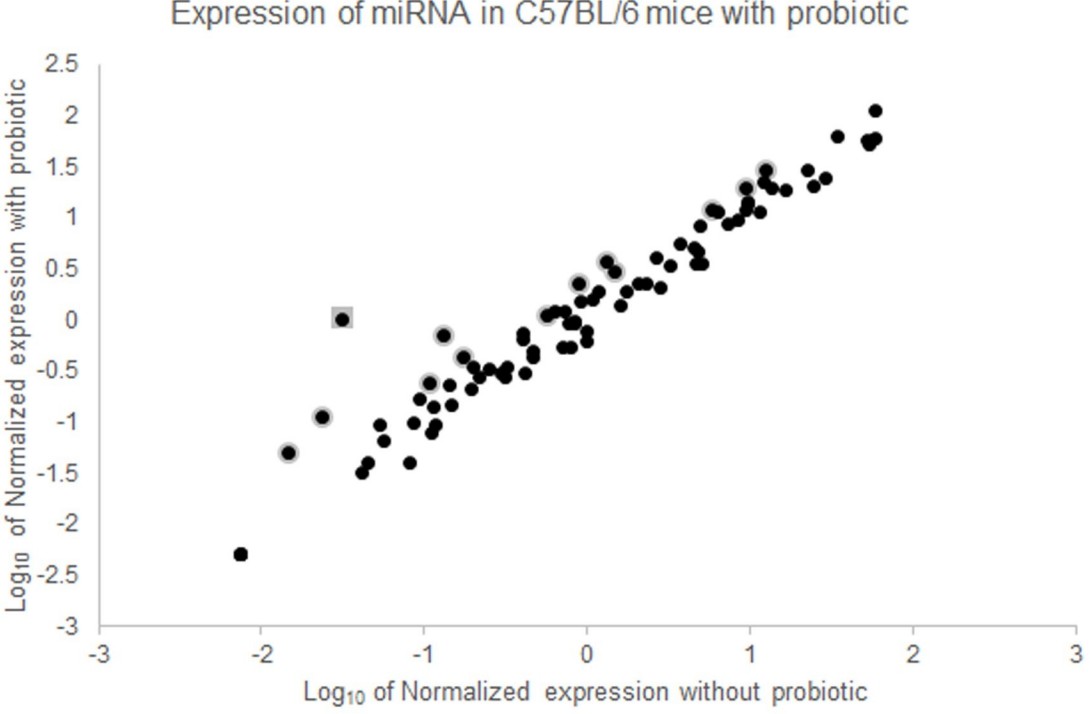

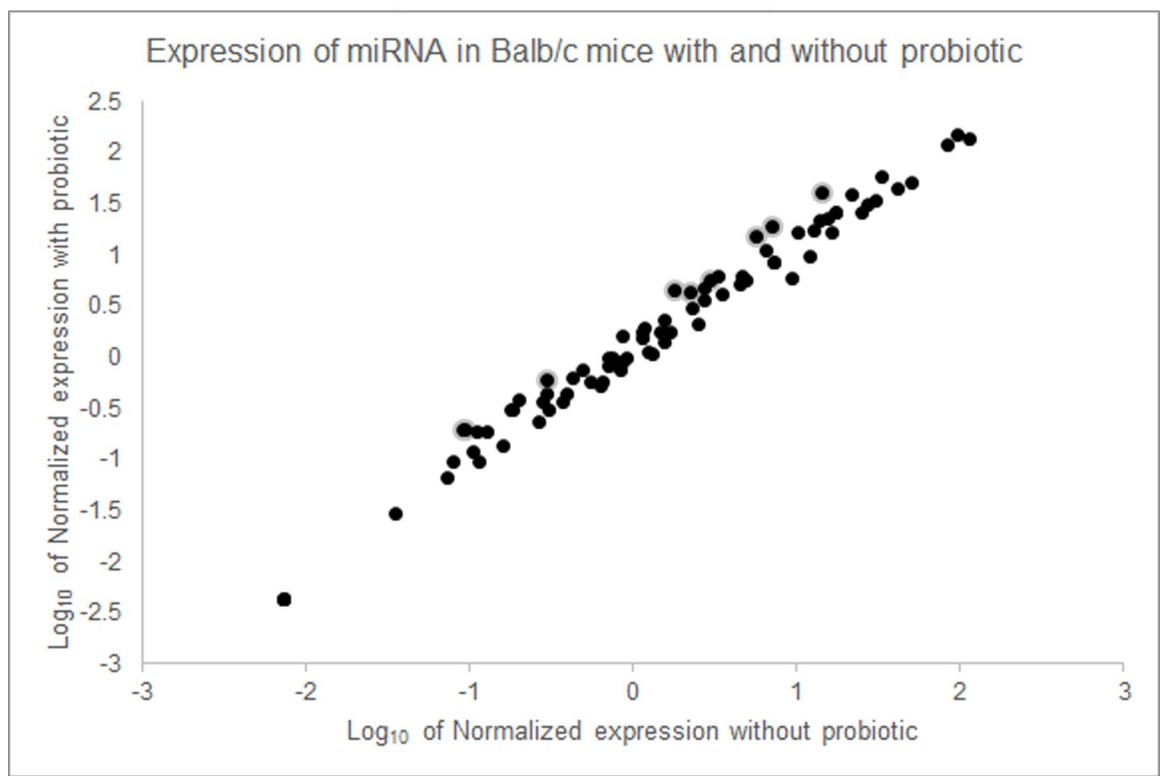

**Fig 2. *B. longum* changes neurologically relevant miRNA expression in both Balb/c and C57BL/6 mice.** Mice that have been fed $10^9$ *B. longum* for three weeks show changes in expression of some neurologically relevant miRNAs in the brain. PCR arrays of miRNAs from the brains of C57BL/6 (top) and Balb/c (bottom) mice reveal that the bacteria does seem to influence gene expression of miRNAs in the brain. Those with greater than 2-fold difference in expression are highlighted in grey. mmu-mir-652-3p is marked with a square in the top graph (see below).

**Table 2. miRNAs that are at least 1.5-fold different in mice that are fed *B. longum* versus those that are fed only media.**

| Gene | Balb/c change in expression | C57BL/6 change in expression |
|---|---|---|
| mmu-let-7b-5p | 2.79 | 2.18 |
| mmu-let-7e-5p | 1.54 | 2.4 |
| mmu-miR-130a-3p | 1.92 | 1.95 |
| mmu-miR-181a-5p | 1.87 | 2.12 |
| mmu-miR-181c-5p | 2.00 | 1.54 |
| mmu-miR-20b-5p | 1.56 | 5.63 |
| mmu-miR-298-5p | 1.75 | 1.86 |
| mmu-miR-29c-3p | 1.56 | 1.93 |
| mmu-miR-320-3p | 1.54 | 2.52 |
| mmu-miR-346-5p | 1.95 | 2.3 |

All genes listed showed a change in both mouse strains tested.

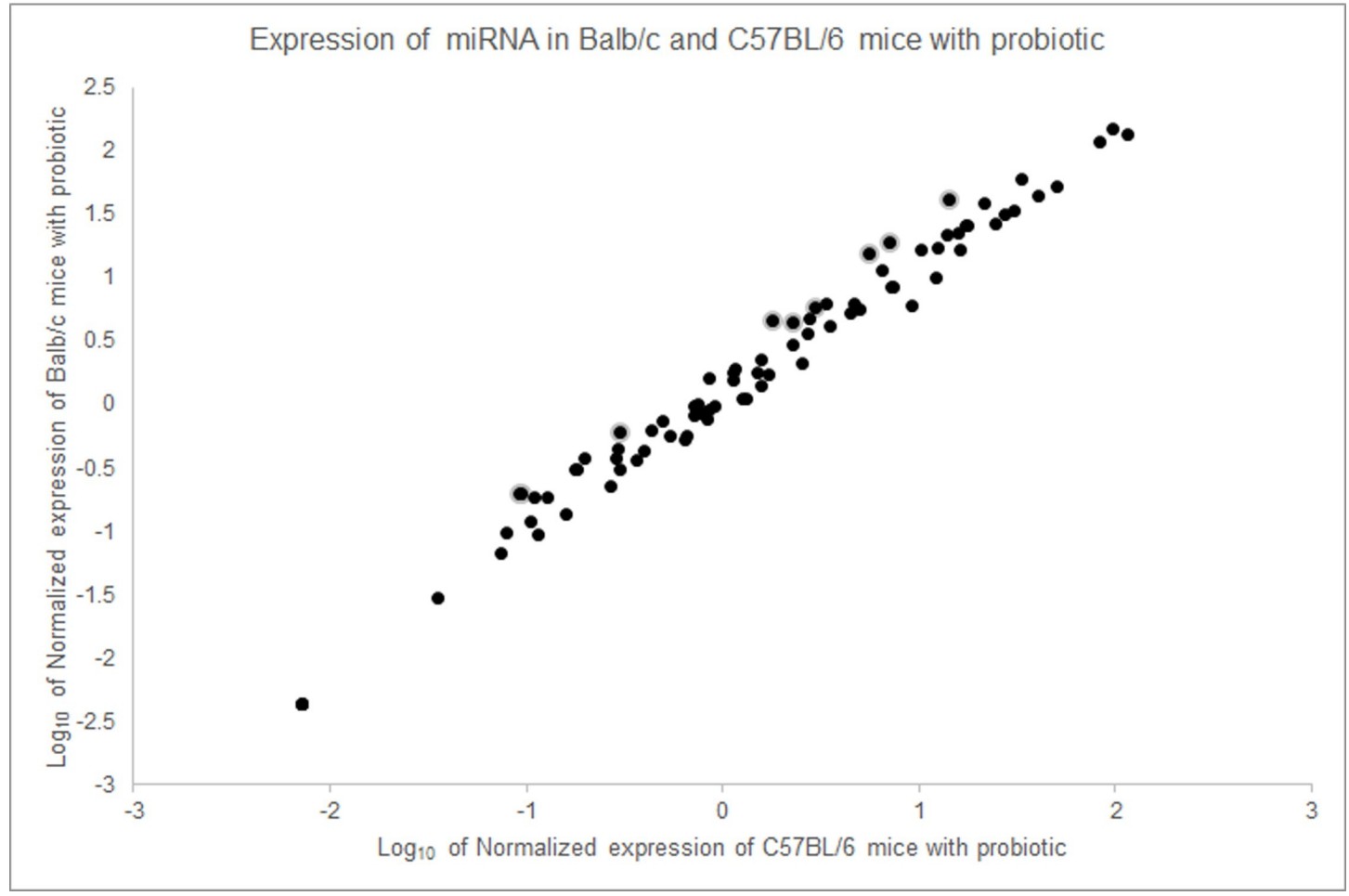

**Fig 3. Expression of miRNA in the brains of mice with *B. longum* treatment.** Expression of brain miRNAs were measured using a PCR array screening of miRNAs associated with neurological development and health. Without treatment, some miRNAs, including mmu-mir-652-3p, are expressed differently in Balb/c and C57BL/6 mice (see Fig 1). With treatment, expression of most miRNAs tested align between the two strains of mice.

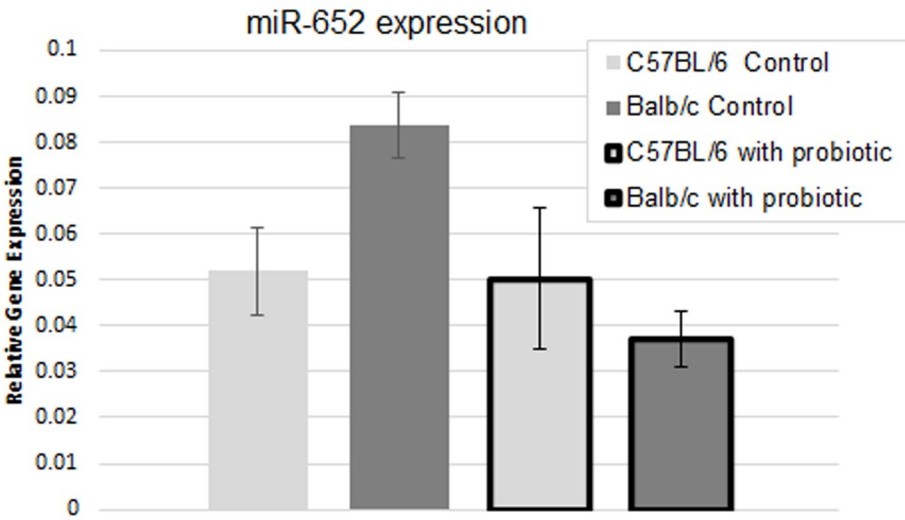

**Fig 4. Expression of mmu-mir-652-3p in the brains of mice with and without *B. longum* treatment.** Expression of mmu-mir-652-3p in the brains of Balb/c and C57BL/6 mice was measured using a qPCR. Values are relative to Hs-RNU6-2 and error bars represent one standard deviation. The change in expression in the Balb/c mice is statistically significant ($p < 0.05$). n = 4 per bar.

To confirm the difference in mmu-mir-652-3p expression upon treatment with *B. longum*, qPCR was performed, testing expression of the single miRNA (Fig 4). Three-week treatment with *B. longum* significantly decreased expression of mmu-mir-652-3p in Balb/c mice, but not C57BL/6 mice. This change was unexpected given the array results, but the final conclusion is the same. These results indicate that *B. longum* could normalize expression of developmentally important miRNAs like, mmu-mir-652-3p, between Balb/c and C57BL/6 mice.

## Consequences of *B. longum* induced miRNA levels

In order to evaluate the consequences of the different expression level of these miRNAs with *B. longum*, we measured the expression of *Dab1*, Disabled-1 (NCBI Gene ID: 13131), an mRNA targeted by mmu-mir-652-3p. *Dab1* is involved in cell positioning and higher order brain structure during fetal development. Based on regulatory RNA activity, changes in the expression levels of miRNA are likely to influence gene expression of the mRNAs that they target. We used qPCR to measure the levels of *Dab1* mRNA. In C57BL/6 mice, which according to the PCR data have no discernable differences in expression of mmu-mir-652-3p upon treatment, *Dab1* seems to have no difference in expression levels with or without treatment with *B. longum*. In contrast, in Balb/c mice, which show a marked decrease in expression of the repressor mmu-mir-652-3p in qPCR, there is a corresponding increase in the levels of *Dab1* mRNA (Fig 5). These results are consistent with the hypothesis that feeding mice *B. longum* causes a decreased expression of the repressor mmu-mir-652-3p which permits higher levels of *Dab1* mRNA.

## General conclusion

As the profound interactions between the gut microbiota and the structure, chemistry, and activity of the brain continue to be discovered, the functional details of how the microbiota-gut-brain-axis works are starting to be resolved. Previous studies have suggested *B. longum* can cause changes in brain chemistry and activity [20]. Other research has revealed that

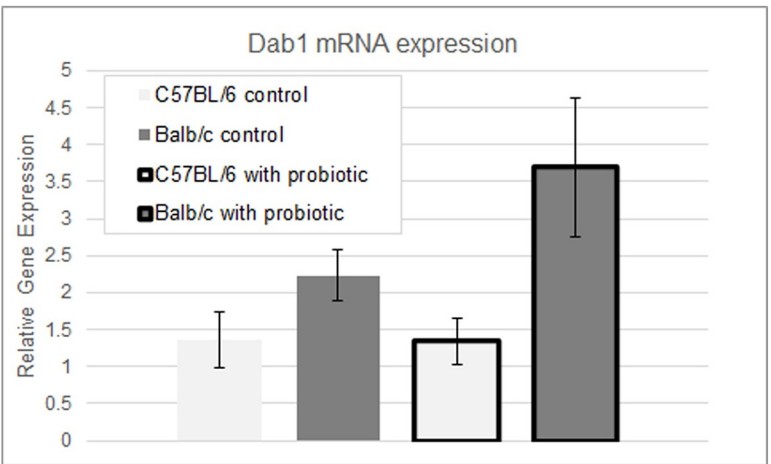

**Fig 5. *Dab1* mRNA levels in the brains of mice with and without *B. longum* treatment.** Expression of *Dab1* in the brains of Balb/c and C57BL/6 mice was measured using a qPCR. Values are relative to *Rer1* control and error bars represent one standard deviation. The change in expression in Balb/c mice is statistically significant (p<0.05). n = 4 per bar.

modifications to expression of Dab1 early in life can have cause lasting changes in behavior of adult mice [21]. Our study, while based on a small sample size, adds further evidence that digestive probiotics can influence miRNAs that in turn regulate expression of important neurological mRNAs. Our data are consistent with a mechanism in which *B. longum* decreases expression of miRNA 652-3p (by a process that has not yet been revealed) and the reduced levels of miRNA 652-3p lead to increased mRNA stability of Dab1 which in turn changes behavior in adult Balb/c mice. There is still much to be clarified about the complex interactions of the microbiota-gut-brain axis, but the activity of miRNAs could be one mechanism by which gut microbes are able to impact brain activity.

## Supporting information

**S1 Data.**
(XLS)

**S1 Table. PCR array data.** This table contains the complete data set from the PRC arrays.
(PDF)

## Author Contributions

**Conceptualization:** Cara Horstman, Clayton Carlson.

**Data curation:** Laura DeVries, Cara Horstman, Marie Fossell.

**Formal analysis:** Laura DeVries, Cara Horstman, Marie Fossell, Clayton Carlson.

**Investigation:** Laura DeVries.

**Methodology:** Laura DeVries, Cara Horstman, Clayton Carlson.

**Project administration:** Clayton Carlson.

**Supervision:** Clayton Carlson.

**Writing – original draft:** Clayton Carlson.

**Writing – review & editing:** Laura DeVries, Cara Horstman, Marie Fossell, Clayton Carlson.

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
