## [Decision Letter · Decision Letter 0]

24 Feb 2021

PONE-D-20-38290

Ingestion of *Bifidobacterium longum* changes miRNA levels in the brains of mice

PLOS ONE

Dear Dr. Clayton,

Thank you for submitting your manuscript to PLOS ONE. After careful consideration, we feel that it has merit but does not fully meet PLOS ONE’s publication criteria as it currently stands. Therefore, we invite you to submit a revised version of the manuscript that addresses the points raised during the review process.

A major revision decision has been made for your article in line with the referee comments. The referee reports are attached. Your letter can be re-evaluated after making the necessary arrangements as a result of the referee comments.

Please submit your revised manuscript by 10 March. If you will need more time than this to complete your revisions, please reply to this message or contact the journal office at plosone@plos.org. Please include the following items when submitting your revised manuscript:

We look forward to receiving your revised manuscript.

Kind regards,

Gulcin Avci

Academic Editor

PLOS ONE

2. Please include captions for your Supporting Information files at the end of your manuscript, and update any in-text citations to match accordingly. Please see our Supporting Information guidelines for more information: http://journals.plos.org/plosone/s/supporting-information

**Comments to the Author**

1. Is the manuscript technically sound, and do the data support the conclusions?

Reviewer #1: Partly

Reviewer #2: Yes

2. Has the statistical analysis been performed appropriately and rigorously? 

Reviewer #1: Yes

Reviewer #2: Yes

3. Have the authors made all data underlying the findings in their manuscript fully available?

Reviewer #1: No

Reviewer #2: Yes

4. Is the manuscript presented in an intelligible fashion and written in standard English?

Reviewer #1: Yes

Reviewer #2: Yes

5. Review Comments to the Author

Reviewer #1: There are several minor comments, which should be addressed in order to improve quality of the paper:

Line 32: please add the references which studies demonstrated these mechanism

Line 36: “Studies of blood metabolites…” but you add only a reference. Please revise it.

Line 154-156: “These data suggest that in addition to the reported behavioral differences between these two strains of mice, there are also differences in expression of regulatory RNAs in the brain [12] Is this your results or Savignac et al. 2014? It is not understandable. Please revişe it.

The discussion section in this study is very poor. Author should be revised the discussion section extensively.

Author used the only fifteen literatures in the manuscript, because they did not discuss their results with other studies. The authors need to do a better job at explaining the necessity of this work and the importance of their conclusions. Presenting of the results and the discussion is generally poor.

Reviewer #2: Dear Authors,

Thank you for your interesting manuscript.

please answer my questions and concerns regarding the study:

1. How can you confirm that the bowel flora of the mice was the same the commencing stage?

2. Why mir-652, was the selected mir? Can you please give a reference for this decision?

3. Did you prove the level of microbiota in the mice bowel before and after trial? The level could be different in each mouse before and after trial and it could affect the results

6. PLOS authors have the option to publish the peer review history of their article (what does this mean?). If published, this will include your full peer review and any attached files.

Reviewer #1: No

Reviewer #2: **Yes: **Shahram Agah

---

## [Author Response · Author response to Decision Letter 0]

1 Mar 2021

We have addressed all the comments from the editor and two reviewers.

---

## [Editor Report · Decision Letter 1]

26 Mar 2021

Ingestion of *Bifidobacterium longum* changes miRNA levels in the brains of mice

PONE-D-20-38290R1

Dear Dr. Clayton Carlson,

We’re pleased to inform you that your manuscript has been judged scientifically suitable for publication and will be formally accepted for publication once it meets all outstanding technical requirements.

Kind regards,

Gulcin Avci

Academic Editor

PLOS ONE

---

## [Editor Report · Acceptance letter]

5 Apr 2021

PONE-D-20-38290R1 

Ingestion of *Bifidobacterium longum* changes miRNA levels in the brains of mice 

Dear Dr. Carlson:

I'm pleased to inform you that your manuscript has been deemed suitable for publication in PLOS ONE. Congratulations! Your manuscript is now with our production department. 

Kind regards, 

on behalf of

Dr. Gulcin Avci 

Academic Editor

PLOS ONE